# High-Efficiency Single-Cell Containment Microdevices Based on Fluid Control

**DOI:** 10.3390/mi14051027

**Published:** 2023-05-11

**Authors:** Daiki Tanaka, Junichi Ishihara, Hiroki Takahashi, Masashi Kobayashi, Aya Miyazaki, Satsuki Kajiya, Risa Fujita, Naoki Maekawa, Yuriko Yamazaki, Akiko Takaya, Yuumi Nakamura, Masahiro Furuya, Tetsushi Sekiguchi, Shuichi Shoji

**Affiliations:** 1Research Organization for Nano & Life Innovation, Waseda University, 513 Waseda Tsurumakicho, Shinjuku-ku, Tokyo 162-0041, Japant.sekiguchi@ruri.waseda.jp (T.S.); 2Medical Mycology Research Center, Chiba University, 1-8-1 Inohana, Chuo-ku, Chiba 260-0856, Japan; jishihara@chiba-u.jp (J.I.); hiroki.takahashi@chiba-u.jp (H.T.); akiko@faculty.chiba-u.jp (A.T.); 3Molecular Chirality Research Center, Chiba University, 1-33 Yayoi-cho, Inage-ku, Chiba 263-8522, Japan; 4Plant Molecular Science Center, Chiba University, 1-8-1 Inohana, Chuo-ku, Chiba 260-8675, Japan; 5School of Fundamental Science and Engineering, Waseda University, 3-4-1 Okubo, Shin-juku-ku, Tokyo 169-8555, Japan; maajiired@gmail.com (M.K.); miyazaki_aya@toki.waseda.jp (A.M.); kajiya.satsuki@gmail.com (S.K.); shojis@waseda.jp (S.S.); 6Department of Natural Products Chemistry, Graduate School of Pharmaceutical Sciences, Chiba University, Chiba 260-8675, Japan; maekawanaokii99@gmail.com; 7Department of Dermatology, Osaka University Graduate School of Medicine, Osaka 565-0871, Japan; y.yamazaki@derma.med.osaka-u.ac.jp (Y.Y.); ymatsuoka@derma.med.osaka-u.ac.jp (Y.N.); 8Department of Dermatology, Chiba University Graduate School of Medicine, Chiba 260-8670, Japan; 9Cutaneous Allergy and Host Defense, Immunology Frontier Research Center, Osaka University, Osaka 565-0871, Japan; 10Graduate School of Advanced Science and Engineering, Waseda University, 3-4-1 Okubo, Shinjuku-ku, Tokyo 169-8555, Japan; mfuruya@waseda.jp

**Keywords:** microfluidic devices, cell culture, single cell, fluid control

## Abstract

In this study, we developed a comb-shaped microfluidic device that can efficiently trap and culture a single cell (bacterium). Conventional culture devices have difficulty in trapping a single bacterium and often use a centrifuge to push the bacterium into the channel. The device developed in this study can store bacteria in almost all growth channels using the flowing fluid. In addition, chemical replacement can be performed in a few seconds, making this device suitable for culture experiments with resistant bacteria. The storage efficiency of microbeads that mimic bacteria was significantly improved from 0.2% to 84%. We used simulations to investigate the pressure loss in the growth channel. The pressure in the growth channel of the conventional device was more than 1400 PaG, whereas that of the new device was less than 400 PaG. Our microfluidic device was easily fabricated by a soft microelectromechanical systems method. The device was highly versatile and can be applied to various bacteria, such as *Salmonella enterica* serovar Typhimurium and *Staphylococcus aureus*.

## 1. Introduction

The cultivation and analysis of bacteria (single cells) using microfluidic devices were studied extensively. Grünberger et al. [1] and Bai et al. [2] captured and cultured single cells on a single-micron scale using microelectromechanical systems technology. There are various types of microfluidic devices for cell culturing [3], including microfluidic chamber and bacterial enrichment devices for cultivating bacteria. For example, Probst et al. performed single-cell analysis by combining a microfluidic device with laser tweezers. They relocated filamentous *Escherichia coli* WT (MG1655) from its growing microcolony by laser manipulation [4]. Grünberger et al. used microfluidic devices to analyze the growth patterns of bacteria in microcolonies [5]. Kim et al. developed a microfluidic device that traps and accumulates cells by using a concentrator array with an arrowhead-like ratchet structure next to the device channel [6]. Applications in bacterial culture analysis, in which examining single cells is important, were also developed. Bell et al. established a technique for trapping cells in specific culture arrays using microfluidic devices with fluidic control that minimized the effect on the growth rate. During fluorescence imaging, cells are held over several hours and can continue to grow and divide within the device [7]. Leea and Hunga developed a microfluidic device for selective trapping of cell pairs and simultaneous optical characterization that could bring two cell populations into membrane contact using an array of trapping channels with a 2 × 2 μm cross section [8]. Moffitt et al. fabricated an agarose-based microfluidic device for cell culture. Densely patterned tracks allowed hundreds of thousands of cells to be cultivated and imaged on a single agarose pad over 30–40 generations, which drastically increased single-cell measurement throughput [9]. Guo et al. presented a simple, inexpensive method for the capture and release of bacteria contained in an array of conical nanopores on a membrane in a microfluidic device. The device also specifically captured cyanobacteria from a mixed suspension of cyanobacteria and chlamydomonas with a selectivity as high as 90% [10]. Probst et al. trapped single cells by placing structures in a microfluidic device channel and reported that the technique was suitable for trapping and analyzing a single, small bacterium [11]. Priest et al. developed a bacterial culture device with a pattern size of 0.6 μm using an electron beam lithography system, and their device was used to acquire and analyze a single-cell dataset of ~10,000 cells across 24 samples quantitatively in under 1 h [12].

The analysis of single cells using a mother machine microfluidic device attracted much attention [13,14]. The mother machine device for trapping single cells has a channel width of 1 μm and requires high-tech microfabrication. Moolman et al. used electron beam lithography to develop microfluidic devices with 0.3 to 0.8 μm growth channels that could load and image Gram-positive and Gram-negative bacteria over a number of generations [15]. Yang et al. fabricated culture channels with a channel width of 0.6–1.0 μm by combining electron beam lithography and photolithography, and the microfluidic device could cultivate single-cell *E. coli*, enabling the effect of the channel dimensions on maternal cell growth to be determined [16]. These mother machine microfluidic devices were used in various single-cell analyses and bacterial division studies [17,18]. Nakaoka and Wakamoto obtained large-scale single-cell lineage data by time-lapse microscopy with a microfluidic device and demonstrated that no replicative aging occurred in old-pole cell lineages of *Schizosaccharomyces pombe* cultured under constant favorable conditions [19]. Cama et al. combined a mother machine microfluidic device and time-lapse auto-fluorescence microscopy to quantify antibiotic accumulation in hundreds of individual *E. coli* cells rapidly [20]. Kaiser et al. integrated a mother machine microfluidic device with Mother Machine Analyzer software, and the system achieved high accuracy in cell segmentation and tracking and the leveraged editing procedure allowed streamlined high-throughput curation [21]. Tsuneda et al. also conducted bacterial culture experiments with conventional devices (mother machine type) [22]. Mother machine microfluidic devices are useful for applications such as studying bacterial chemotaxis. However, these devices cannot trap bacteria well in the growth channel, and thus, to achieve this, the device is centrifuged, tilted, or shaken, which risks damaging or stressing the bacteria. Microfluidic devices with a drain structure that allows bacteria to enter the growth channel efficiently were developed. Spivey et al. used a 3D printer to fabricate a mother machine microfluidic device with a drain in the culture channel to allow efficient cell capture [23]. Rowat et al. developed a microfluidic device to track multiple lineages in parallel by trapping single cells and constraining them to grow in lines for as many as eight divisions, which allowed phenotypes to be correlated with the age and genealogy of single cells over time [24]. Mother machine microfluidic devices with these drain structures are difficult to fabricate because the fabrication process is complex and it is difficult to make devices with a channel width of less than 1 μm. In addition, the devices are structurally prone to damage.

Mother machine microfluidic devices are useful for applications such as studying bacterial chemotaxis. However, these devices cannot trap bacteria well in the growth channel, and thus, to achieve this, the device is centrifuged, tilted, or shaken, which risks damaging or stressing the bacteria. In this study, we succeeded in developing a microfluidic device that can be stored in a growth channel without stressing the bacteria (Figure 1).

## 2. Materials and Methods

### 2.1. Device Design

Figure 2 shows a schematic of the high-efficiency bacterial capture device developed in this study. Figure 2a shows an overall view of the microfluidic device. The device had 1000 branching channels in which cells were trapped. The spacing between each growth channel was 2 μm. Figure 2b shows the dimensions of the device, which had a three-stage structure, at heights of 0.4, 1.2, and 10 μm from the bottom. Figure 2c shows an overview of the bacterial capture method. Bacteria flowing in the main channel were carried through the device by fluid flowing into the growth channel, throttle channel, and side channels.

### 2.2. Device Fabrication

The microfluidic devices were fabricated using a three-stage exposure method.

Figure 3 shows the fabrication process for the microfluidic device. The SU-8 negative photoresist (SU-8 3010 and SU-8 2000.5, Kayaku MicroChem, Tokyo, Japan) was spin-coated on a silicon substrate. The SU-8-coated substrate was patterned with a UV exposure machine (MA/BA6, SUSS MicroTec, Munich, Germany) and developed with SU-8 developer to produce an SU-8 mold. The PDMS resin and curing agent (SILPOT 184, Dow Corning Toray, Tokyo, Japan) were mixed in a 10:1 ratio and poured onto the SU-8 mold. After degassing, the mixture was cured at 75 °C for 1 h, and then, the cured PDMS was demolded. High-aspect-ratio flexible parallel PDMS walls were obtained. The PDMS was bonded directly to an O_2_ plasma-treated glass substrate.

### 2.3. Fluid Simulation

Three-dimensional CFD simulations were performed to obtain pressure distribution for the device reported by Spivey et al. [23] and our device. The momentum field was described by the mass continuity and Navier–Stokes equations for incompressible fluids with constant density and viscosity. The set of equation was solved with a commercial CFD code, Simcenter Star-CCM+ version 2021.1. The width and height of the growth channel were 2.0 μm and 1.2 μm, respectively. In the conventional device design, the throttle channel narrows horizontally (Figure 4b), but in our device, the channel was stepped in the height direction to trap cells (Figure 4a). All the channels were assumed non-slip condition.

### 2.4. Bacterial Culture and Fluid Experiments

#### 2.4.1. Bacterial Strains and Culture Conditions

The mutant strain of *S. aureus*, pFK55/LAC strain, was constructed by electroporation to recombine the plasmid pFK55 [25] into the LAC strain [26]. The electroporation was conducted as described in Monk et al. [27]. In detail, the overnight cultures of *S. aureus* (LAC strain) were grown in 4 mL brain heart infusion broth (BHI) (in 14 mL tubes) with horizontal shaking (180 rpm) using StackShake (WAKO). After being harvested by centrifugation at 4000× *g* for 3 min, the cells were resuspended in an equal volume of autoclaved ice-cold water. The cells were then repeatedly centrifuged and resuspended first in 1/3rd, in 1/12th, 1/230th, and, finally, in 1/300th the volume of autoclaved ice-cold 10% (*w*/*v*) glycerol. Aliquots of 100 µL were frozen at −80 °C. For electroporation, the cells were thawed on ice for 5 min and were then left at room temperature for 5 min before undergoing centrifugation (4000× *g* for 3 min), followed by resuspension in 50 µL of 10% glycerol and 500 mmol/L sucrose (filter sterilized). After being incubated with the plasmid DNA (1 µg) on ice for 10 min, the sample was transferred to a 2 mm electroporation cuvette (Bio-Rad) at room temperature, and pulsed at 9 kV per cm. The cells were incubated in 1 mL of BHI supplemented with 500 mmol/L sucrose (filter sterilized) at 37 °C for 2 h before plating on BHI agar containing 10 µg of chloramphenicol and incubated at 37 °C overnight for CFU count. On the other hands, *Salmonella* strain used in this study was the derivative of *Salmonella enterica* serovar 500 Typhimurium χ3306 [28]. To culture the *S. aureus* and *S. enterica*, we used LB and MHB liquid media, respectively. For subsequent experiments, the *S. aureus* and *S. enterica* were grown overnight in each medium at 37 °C with vigorous shaking.

#### 2.4.2. Procedure for Microfluidic Experiments

Bacterial cells were diluted to the OD_595_ value of 1.0 before loading the cells into the device. To trap the cells in the growth channels, we mounted the device on the microscope stage. To load the cells into the inlet, we used 30 mL syringes (Terumo) driven by a syringe pump (As One) at a rate of 240 μL/h. The media syringe and the fluidic inlet were connected by a silicon tube with a 2 mm inner diameter. We used Teflon tubes to interface between the silicon tubes with 2 mm and 4 mm inner diameter inserted into the fluidic inlet. During the observation, the device was constantly supplied with a fresh medium from the inlet. All data were acquired using a DS-Fi3 camera sets attached to Nikon ECLIPSE Ts2R inverted microscope (Nikon, Tokyo, Japan). The microscope was equipped with an electric microscope stage (Prior Scientific KK, Cambridge, UK) to exchange the sample position during the observation. Nikon NIS-Elements AR software (ver. 5.20.02) controlled the camera, motorized stage, and fluorescent and brightfield shutters. To observe the *S. aureus* and *S. enterica* in the growth channels, we used 100× Oil Plan Apo (NA 1.45) and 60× Plan Apo (NA 0.95) objective lens (Nikon), respectively. GFP was monitored using a C-LEDFL470 LED light source (Nikon) and 470 nm C-LED filter (Nikon). Exposure times were adjusted to keep exposure light levels constant between experiments.

## 3. Results and Discussion

### 3.1. Computational Fluid Dynamics Model and Simulations

Computational fluid dynamics (CFD) simulations were performed to evaluate the device geometry on the pressure loss. Figure 4 shows CFD domains: (a) the conventional design, (b) the device designed by Spivey et al. [23], and (c) our device. In the conventional device design, the growth channel is a dead end, and so, there is almost no flow into the growth channel and bacteria cannot be stored; thus, bacteria are forced into the growth channel by centrifugation or overculturing. Spivey et al. [23] attempted to contain bacteria by installing a throttle channel, but the pressure loss in the throttle channel was large.

Figure 5 shows the estimated pressure distribution along streamwise direction. The magnitude of pressure is represented by color in Figure 5a and by the graph in Figure 5b. The higher the pressure, the more difficult it becomes for bacteria to enter the growth channel. Figure 5a,b show that the pressure applied to the growth channel in the conventional device was 1400 PaG, whereas it was about 400 PaG in our device because narrowing the throttle channel in the longitudinal direction reduced the pressure in the growth channel. In conventional devices, centrifugation or excessive incubation is used to force bacteria into the growth channel, but these methods damage the cells. In our device, the fluid flow carries bacteria into the growth channel without stress. Our device had a channel width of 1 to 4 μm, and can store bacteria of various sizes, such as *S. aureus* and *S. enterica*, in the growth channel. The pressure loss in our device was reduced by more than one-third compared with the conventional device. In conventional devices, the throttle channel is narrowed in the *xy*-plane, which increases the flow resistance and results in pressure loss, whereas in our device, the throttle channel was narrowed in the *z*-direction but widened in the *xy*-direction, reducing pressure loss substantially.

### 3.2. Fluid Experiments (with Microbeads)

Reagents and solutions were introduced by syringe (1750CX, Hamilton) using a syringe pump (KDS-100, KD Scientific, Holliston, MA, USA). The syringes and device were connected via ethylene tetrafluoroethylene tubing.

Figure 6 shows the result of fluid experiments in which the efficiency of microbead containment was compared between the conventional device and the newly developed device. The experiments were conducted using microbeads of 1 μm in diameter, which is about the same size as bacteria. Microbeads and pure water were placed in syringes, which were then connected to the fluidic device via Teflon tubing. Microbeads were introduced into each of devices for 300 s. Figure 6a shows the results for our newly developed microfluidic device. The storage efficiency of microbeads reached 80%. Figure 6b shows a conventional device with the same channel depth and width as (a). The storage efficiency of microbeads was less than 1%.

Figure 7 shows the experimental results of containment control using microbeads (size: 1 μm) that mimic cells. Figure 7a shows our device. One microbead was stored in the growth channel about 15 s after the start of the fluid experiment. When focusing on the growth of a single cell, the fluid was stopped in this state. After 60 s, two microbeads were stored in the growth channel. The number of cells stored in the growth channel can be controlled by the length of the inflow time. For example, when trapping a single cell in a growth channel, a single cell can be stored if the flow was stopped in 15 to 30 s. On the other hand, Figure 7b shows a microbead storage experiment with a conventional device. Two microbeads were retracted within 5 s, after which no microbead retraction was observed, even after 300 s had elapsed. There were 1000 growth channels, but only stored two microbeads.

Figure 8 is a graph of microbead storage efficiency. There was no significant difference between our device and the conventional device after 5 s of fluid experiment. However, while conventional devices showed no change from a 0.2% storage efficiency, our device showed a storage efficiency of 20% after 15 s, 50% after 30 s, and 84% after 60 s. As the graph shows, efficient microbead (Mimics cells) storage was achieved by controlling the fluid.

### 3.3. Fluid Experiments (with Bacteria)

Figure 9 shows the results of fluid experiments using *Salmonella enterica* serovar Typhimurium and *Staphylococcus aureus*. Bacteria and culture medium were placed in syringes, which were then were connected to the fluidic device using Teflon tubing. The bacteria were introduced into the fluidic device by pushing the syringe manually. Figure 9a shows the results of fluid experiments using *S. aureus*. The bacteria, which had a diameter of about 1 μm, were stored in the growth channel efficiently. Figure 9b shows a microscope image of a fluid experiment using *S. enterica*, which were oval and had dimensions of about 1 μm by 2–5 μm. *S. enterica* were stored in the growth channel within 10 s of the start of the experiment. *S. aureus* and *S. enterica* stored in the growth channel could be cultured as usual. Both *S. aureus* and *S. enterica* were stored in the growth channel with a probability of more than 90%. In the conventional device, the efficiency was less than 1%, and so, the bacteria that happened to be stored in the growth channel were used for observation. In the conventional method, *S. aureus* could not be stored in the growth channel, and thus, the bacteria were forced into the channel using a centrifuge. In addition, *S. enterica* was stored in the growth channel by swimming after 3 h of preculture. With the device developed in this study, bacteria were stored in the growth channel within several seconds of the fluid being introduced, allowing for efficient culture experiments. Furthermore, there was no damage to the cells because no centrifuge was used. The number of cells stored in the growth channel depended on the number (concentration) of bacteria in the culture medium introduced, and could be stored one at a time. However, the *S. aureus* cells were in clusters, and so, pre-treatment was necessary to separate the cells. Our device allowed cells trapped in the growth channel to remain in the channel, making it possible to the culture medium and introduce antibiotics, and thus, the device is ideal for experiments such as elucidating persistent phenomena.

## 4. Conclusions

We designed and fabricated a microfluidic device for trapping and culturing single bacteria. We trapped a variety of bacteria by adding a longitudinally narrowed throttle channel at the end of the growth channel. In an experiment of storing microbeads that imitate cells, our device achieved 84% storage efficiency, compared to 0.2% for conventional devices. Bacteria were stored in more than 90% of the growth channels, and bacterial culture was possible. The stored bacteria experienced little stress, which will make it possible to accurately perform experiments such as elucidating persistent phenomena. We simulated the pressure loss in the growth channel, and the pressure loss in our device was less than one third that of conventional devices, indicating that various bacteria could be stored efficiently in the growth channel. Our device is highly versatile and could be used with a variety of bacteria in addition to *S. enterica* and *S. aureus*.

## Figures and Tables

**Figure 1 micromachines-14-01027-f001:**
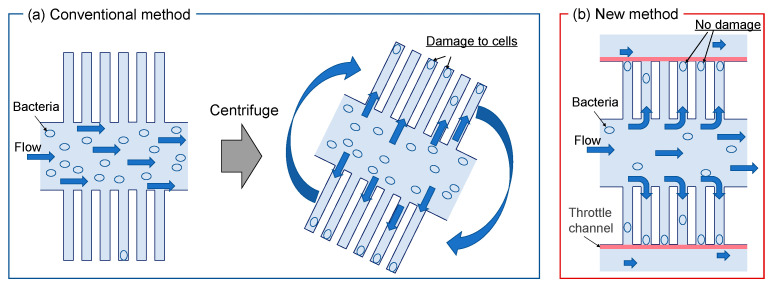
Device concept, (**a**) problems of conventional devices, (**b**) new type of devices with fluid control.

**Figure 2 micromachines-14-01027-f002:**
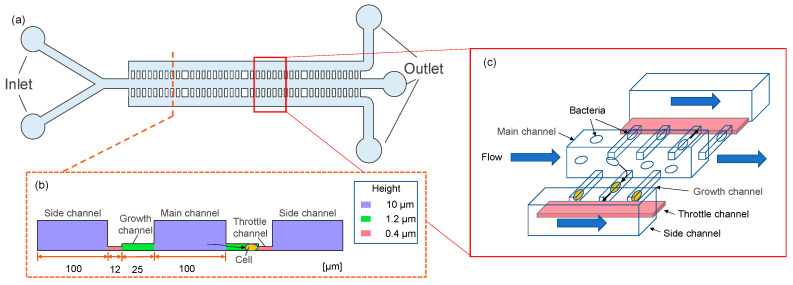
Design of microfluidic devices. (**a**) Branch channel design, (**b**) device dimensions, and (**c**) overview of the bacterial capture method.

**Figure 3 micromachines-14-01027-f003:**
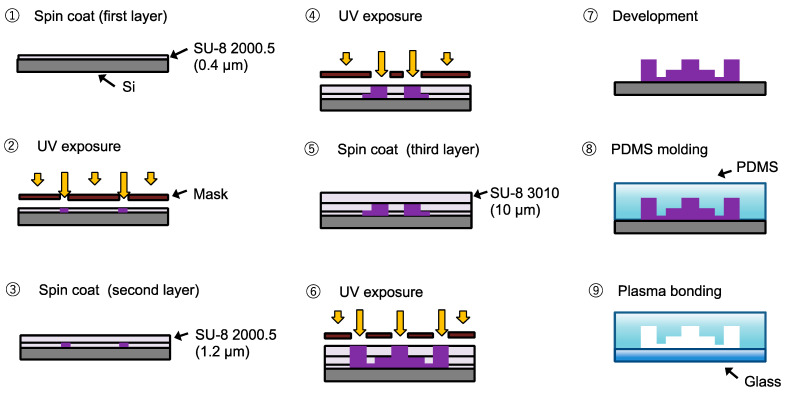
Fabrication process of microfluidic devices.

**Figure 4 micromachines-14-01027-f004:**
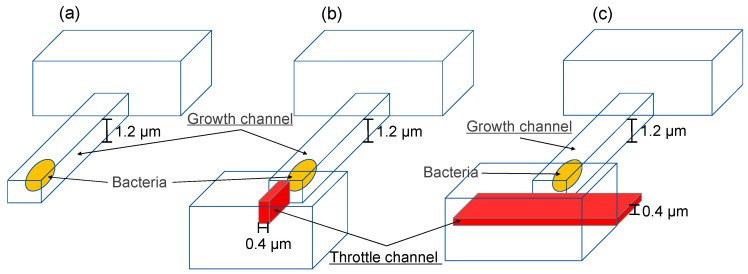
Schematics of the devices used in the simulation. (**a**) Conventional device, (**b**) device reported by Spivey et al. [23] *Anal. Chem.* **2014**, and (**c**) our device.

**Figure 5 micromachines-14-01027-f005:**
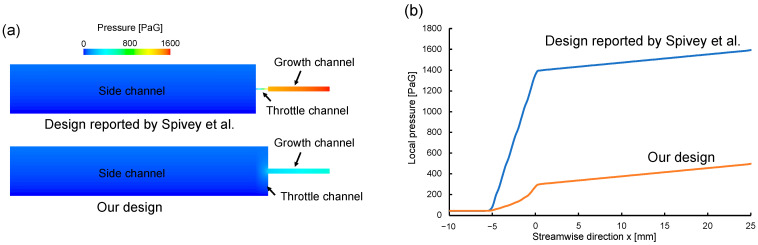
Simulation results of pressure distribution along streamwise direction. (**a**) Pressure distribution from the growth channel to the side channel in the device reported by Spivey et al. [23] (**top**) and our device (**bottom**). (**b**) Pressure in the growth channel in the device reported by Spivey et al. [23] *Anal. Chem.* 2014 and our device.

**Figure 6 micromachines-14-01027-f006:**
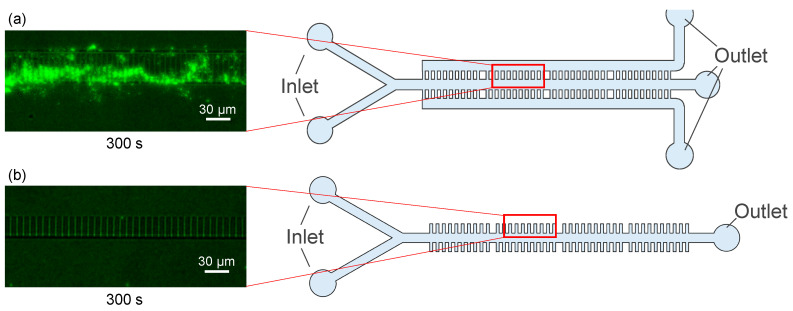
Microscope images showing a comparison of the efficiency of microbead containment. (**a**) Our device, (**b**) conventional device.

**Figure 7 micromachines-14-01027-f007:**
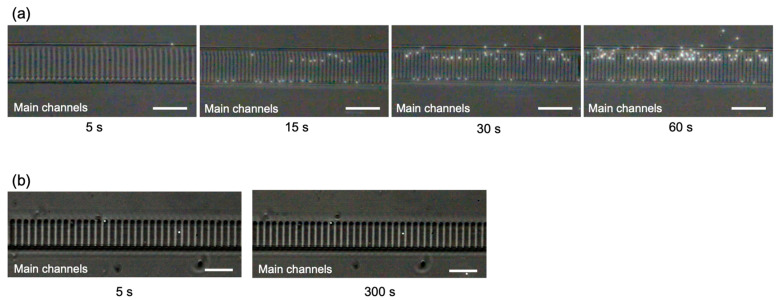
Relationship between the number of microbeads stored and fluid inflow time. (**a**) Our device, (**b**) conventional device. (Bar is 25 μm).

**Figure 8 micromachines-14-01027-f008:**
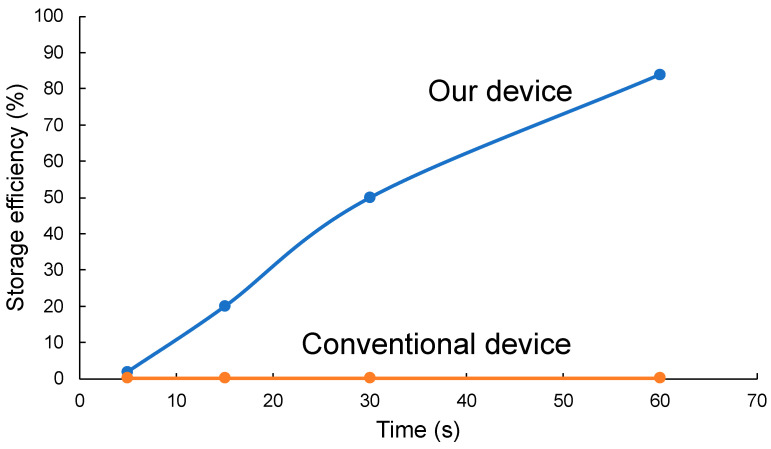
Graph of microbead stored efficiency.

**Figure 9 micromachines-14-01027-f009:**
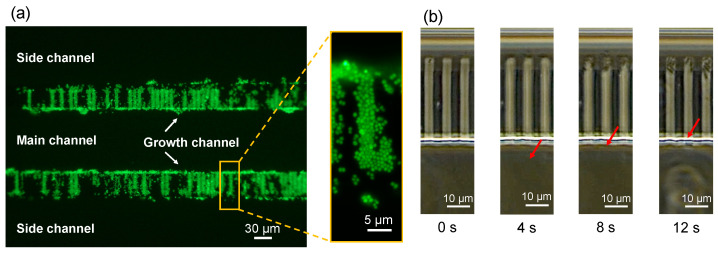
Microscope images of fluid experiments. (**a**) Fluorescence microscope image of the *S. aureus* experiment (**left**) with a magnified image of the growth channel (**right**). (**b**) Optical microscope images of *S. enterica* experiment over time. The red arrows indicate a bacterium moving into the growth channel.

## Data Availability

Not applicable.

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
