# Peer review of "High-Efficiency Single-Cell Containment Microdevices Based on Fluid Control"

_micromachines, 2023, doi:10.3390/mi14051027_

Round 1

Reviewer 1 Report (New Reviewer)

The authors designed and fabricated a microfluidic device for single-cell containment and culture. With the incorporation of side channels and throttle channels, the authors could avoid centrifugation for cell loading which can cause cell damage. While the design is interesting, there are several major problems in the experimentation.

1.       The authors claimed that the new microfluidic device is significantly more efficient than the conventional device. The reported efficiency of the conventional device was around 1 %. There is no description of the usage of the conventional device. If the authors used syringe pumping to process the device, it would be expected that the conventional device would give extremely low yield as it is designed for centrifugation.

2.       The method section is too short. A lot of details are missing. For example, bacteria culture methods; bacterial sample process protocol such as the number of bacteria introduced, and infused flow rates; description of the conventional device; details about simulation such as the software used, the physics, 2D or 3D simulation.

3.       The advantage of the new device was claimed to be giving higher bacterial viability compared with the conventional device. However, the authors didn’t show any data to support this argument.

Author Response

Reviewer 1

The authors designed and fabricated a microfluidic device for single-cell containment and culture. With the incorporation of side channels and throttle channels, the authors could avoid centrifugation for cell loading which can cause cell damage. While the design is interesting, there are several major problems in the experimentation.

Thanks for the constructive feedback.

  1. The authors claimed that the new microfluidic device is significantly more efficient than the conventional device. The reported efficiency of the conventional device was around 1 %. There is no description of the usage of the conventional device. If the authors used syringe pumping to process the device, it would be expected that the conventional device would give extremely low yield as it is designed for centrifugation.

Ans.

When comparing containment efficiencies, it is important that the two experimental conditions be the same. The containment efficiency using a centrifuge varies depending on the number of rotations, making accurate comparisons difficult.

Our device has the advantage of achieving high containment efficiency without the use of a centrifuge.

  1. The method section is too short. A lot of details are missing. For example, bacteria culture methods; bacterial sample process protocol such as the number of bacteria introduced, and infused flow rates; description of the conventional device; details about simulation such as the software used, the physics, 2D or 3D simulation.

Ans.

Methods section added.

Added lines 155-163 and 230-276.

  1. The advantage e of the new device was claimed to be giving higher bacterial viability compared with the conventional device. However, the authors didn’t show any data to support this argument.

Ans.

Thank you for your very helpful comments.

The purpose of this paper is to report that the device design dramatically increases the efficiency of microbeads and bacteria containment. While it is important to quantitatively evaluate the survival rate, we plan to separate the paper on the differences in growth of various bacteria into subsequent papers.

Reviewer 2 Report (New Reviewer)

The authors present a hydrodynamic-based entrapment for bacterium culture in confined microchannels. 

The design is sound and numerical simulation as well as experiment proof were given. 

however I have several questions and concerns.

1. the multistep microchannel design the authors proposed is sound and have been shown to facilitate loading of the bacteria and shown improvement over previous design by Spivey et al. over pressure distribution.

How does the chemical transport of nutrient in compared to authors design and Spivey's design and to those in original mother machines?

especially considering the throttle channel spans for 12 micrometer. Scales much more than any of the mother machines channel 

2. In a previous work, doi:10.3390/mi13040576, as well as in Mother Machine chips, it is not easy to retain bacteria in the channels and have enough perfusion of nutrient to support continuous bacterial outgrowth to study bacterial dynamics. 

I think if authors can add more results seeing bacteria outgrow stably, will be more convincing. 

3. The authors describe device design and fabrication. But failed to mention the details of the bacteria culture & strains, imaging conditions and setup, how S. aureus were to transformed with fluorescence. 

The minimum information necessary for reproducable results will be important

4. what is the spacing between each growth channel?

Although the proposed design have less pressure distribution. But Figure 6A, figure 10A, it looks as if the pdms in between channels were not properly bonded. or else there will be no bacteria spreading, there should be a clear channel boundary. 

Author Response

Reviewer 2

The authors present a hydrodynamic-based entrapment for bacterium culture in confined microchannels.

The design is sound and numerical simulation as well as experiment proof were given.

however I have several questions and concerns.

Thank you for your helpful input.

I have revised the paper.

  1. the multistep microchannel design the authors proposed is sound and have been shown to facilitate loading of the bacteria and shown improvement over previous design by Spivey et al. over pressure distribution.

How does the chemical transport of nutrient in compared to authors design and Spivey's design and to those in original mother machines?

especially considering the throttle channel spans for 12 micrometer. Scales much more than any of the mother machines channel

Ans.

The throttle channel height is as low as 400 nm, making it difficult for highly viscous materials such as PEG to pass through. However, we confirmed that the culture medium used in this study passed through the throttle channel and came out of the outlet of the side channel without any problem, although it was more viscous than water.

  1. In a previous work, doi:10.3390/mi13040576, as well as in Mother Machine chips, it is not easy to retain bacteria in the channels and have enough perfusion of nutrient to support continuous bacterial outgrowth to study bacterial dynamics.

I think if authors can add more results seeing bacteria outgrow stably, will be more convincing.

 Ans.

Thank you for your important comments.

How stored bacteria grow is an important discussion, but the purpose of this paper is with respect to efficient storage in growth channels, and I would like to separate the discussion of bacterial growth rates into a subsequent paper.

  1. The authors describe device design and fabrication. But failed to mention the details of the bacteria culture & strains, imaging conditions and setup, how S. aureus were to transformed with fluorescence.

The minimum information necessary for reproducable results will be important

Ans.

Details of bacterial culture were added.

Added lines 230-276.

  1. what is the spacing between each growth channel?

Although the proposed design have less pressure distribution. But Figure 6A, figure 10A, it looks as if the pdms in between channels were not properly bonded. or else there will be no bacteria spreading, there should be a clear channel boundary.

Ans.

Added note on growth channel spacing.

Added lines 116-117.

The height of the throttled channels was set to 400 nm, taking into account flow resistance and other factors. However, about 1-5% of microbeads (bacteria) enter the throttle channel because PDMS is flexible and deforms when pressure is applied. This is unavoidable with the current materials and device design.

Round 2

Reviewer 1 Report (New Reviewer)

Added lines 155-163 and 230-276 should be placed in the Method Section not the Result and discussion section.

Author Response

Added lines 155-163 and 230-276 should be placed in the Method Section not the Result and discussion section.

Thanks for the constructive feedback.

The relevant part was described in the experimental section.

Reviewer 2 Report (New Reviewer)

I accept author's comments of reply. But for nutrient transport, the authors replied that they observed the medium flow. If there is any time lapse movies and/or numerical simulation with Fick's transport modeled it would have been more convincing and improve the quality of the paper. 

For figure 10. I see the authors revised that the gap between growth channel is 2 micron. in our experience it's too thin. Perhaps that's the reason the PDMS is poorly bonded which led to the leakage between gap and S. aureus go within. 

While brain heart infusion BHI is well known in the microbiology community, it and some of the acronyms in the revised methods section may not to the microfabrication community of Micromachines' readers. Recommend spell checks by editorial office before publishing. 

Author Response

I accept author's comments of reply. But for nutrient transport, the authors replied that they observed the medium flow. If there is any time lapse movies and/or numerical simulation with Fick's transport modeled it would have been more convincing and improve the quality of the paper.

For figure 10. I see the authors revised that the gap between growth channel is 2 micron. in our experience it's too thin. Perhaps that's the reason the PDMS is poorly bonded which led to the leakage between gap and S. aureus go within.

While brain heart infusion BHI is well known in the microbiology community, it and some of the acronyms in the revised methods section may not to the microfabrication community of Micromachines' readers. Recommend spell checks by editorial office before publishing.

Thanks for the constructive feedback. The reviewer's input will be helpful for future research. Thank you.

The paper included the official name of BHI (Line 160-161).

This manuscript is a resubmission of an earlier submission. The following is a list of the peer review reports and author responses from that submission.

Round 1

Reviewer 1 Report

The Authors designed the microfluidic device to improve the mother machine for trapping a single cell. By adding a throttle channel at the end of growth channels, cells or beads were captured into the channel without using a centrifuge. As throttle channels have a low height and a wide width, it was possible to reduce the applied pressure while allowing the fluid to flow through the channel to guide cells. Although it is an interesting idea, the data did not sufficiently support the authors’ claim. For instance, there is no proof of the achievement of single-cell trapping and high efficiency at the same time. Also, the author did not compare their methods to the conventional approach appropriately. For a fair comparison, the author should use the centrifuge to trap cells in the conventional device and see whether there is damage to the cell while their new device was able to avoid such damage. In addition, extensive editing of writing is required. Overall, I would like to not recommend publishing this article. Below are the specific comments that can be used to improve their article when they submit it to other journals.

 - It is required to show the single-cell trapping and quantify the efficiency of filling. Without such data, it is not valid to claim highly efficient single-cell trapping technology. The authors claimed that the storage efficiency of microbeads was ~80%, but it was not single-bead trapping. They also claimed that both bacteria were trapped with a probability higher than 90%, but it was not single-cell trapping.

- The authors did not compare their approach to the conventional method appropriately. With the conventional device, they must use the centrifuge to trap the cell and see whether cells were damaged. This is because the authors claimed that their device has the advantage of preventing damage to cells by not applying force to the cells.

- The authors claimed that it is possible to exchange the culture media while having cells trapped in the channel. However, there is no demonstration to validate it.

- The authors would present the data more quantitatively. It is required to quantify the number of beads/cells per trap. Also, they should quantify the efficiency at the same time. Such quantification must be shown with the plot.

- There were multiple problems in writing. For instance, the first two paragraphs in “2.2 Device Design” are the same except for the first word. The authors wrote the nouns too consecutively (e.g., mother machine microfluidic device, soft microelectromechanical systems method). Also, there were several grammar errors.

Reviewer 2 Report

This manuscript by Tanaka and collaborators introduces a microfluidic device based on the classic mother machine design but with an added “throttle channel” to the back of the growth chambers. This throttle channel has the potential to make cell loading more efficient, and to reduce centrifugation and pressure stress on the cells, a better design especially for the study of environment changes such as antibiotic stress resistance. However, the manuscript feels hastily written: The introduction is a list of studies that are relevant but it lacks a clear question and a description of how the authors sought to answer it, the materials and methods section lacks critical details for other scientists to be able to reproduce the device, and finally the results section lacks details that must be inferred from the figures. 

Most importantly, it is hard to say because of the seemingly low magnification, but the authors appear to be having binding issues between their glass slide and PDMS chip and are getting their fluorescent beads and bacteria a bit all over the device, including in places where they shouldn’t be. This in turn makes it hard to conclude that the better loading that they observe in Figure 6 comparatively to a classic mother machine design is because of the structure of their chip or because the PDMS is simply lifted when they are flushing and then traps them down.

More specific comments:

Lines 96-97:  “Mother machine microfluidic devices with these drain structures are difficult to fabricate because the fabrication process is complex and it is difficult to make devices with a channel width of less than 1 μm.” 

-> Please elaborate as to why this is difficult

The reagents and materials section (2.1) contains very little information. Information from section 2.3 should be in there. In general, methods are poorly described, with a lot of relevant parameters missing for a paper focussing on a microfluidic chip.

Lines 120 - 126 are a copy and paste of lines 108 - 114

Figures 6A-B, Figure 7, and Figure 8 should indicate the objective magnification in the caption.